# Recent Clinical Treatment and Basic Research on the Alveolar Bone

**DOI:** 10.3390/biomedicines11030843

**Published:** 2023-03-10

**Authors:** Sachio Tsuchida, Tomohiro Nakayama

**Affiliations:** Division of Laboratory Medicine, Department of Pathology and Microbiology, Nihon University School of Medicine, 30-1 Oyaguchikamicho, Itabashi-ku, Tokyo 173-8610, Japan; tsuchida.sachio@nihon-u.ac.jp

**Keywords:** periodontal tissue, periodontal disease, alveolar bone, alveolar surgery, alveolar bone resorption, implant therapy

## Abstract

The periodontal ligament is located between the bone (alveolar bone) and the cementum of the tooth, and it is connected by tough fibers called Sharpey’s fibers. To maintain healthy teeth, the foundation supporting the teeth must be healthy. Periodontal diseases, also known as tooth loss, cause the alveolar bone to dissolve. The alveolar bone, similar to the bones in other body parts, is repeatedly resorbed by osteoclasts and renewed by osteogenic cells. This means that an old bone is constantly being resorbed and replaced by a new bone. In periodontal diseases, the alveolar bone around the teeth is absorbed, and as the disease progresses, the alveolar bone shrinks gradually. In most cases, the resorbed alveolar bone does not return to its original form even after periodontal disease is cured. Gum covers the tooth surface so that it matches the shape of the resorbed alveolar bone, exposing more of the tooth surface than before, making the teeth look longer, leaving gaps between the teeth, and in some cases causing teeth to sting. Previously, the only treatment for periodontal diseases was to stop the disease from progressing further before the teeth fell out, and restoration to the original condition was almost impossible. However, a treatment method that can help in the regeneration of the supporting tissues of the teeth destroyed by periodontal diseases and the restoration of the teeth to their original healthy state as much as possible is introduced. Recently, with improvements in implant material properties, implant therapy has become an indispensable treatment method in dentistry and an important prosthetic option. Treatment methods and techniques, which are mainly based on experience, have gradually accumulated scientific evidence, and the number of indications for treatment has increased. The development of bone augmentation methods has contributed remarkably to the expansion of indications, and this has been made possible by various advances in materials science. The induced pluripotent stem cell (iPS) cell technology for regenerating periodontal tissues, including alveolar bone, is expected to be applied in the treatment of diseases, such as tooth loss and periodontitis. This review focuses on the alveolar bone and describes clinical practice, techniques, and the latest basic research.

## 1. Introduction

To maintain healthy teeth, the structures that support the teeth must be healthy. One of the most important of these structures is the alveolar bone (Figure 1). Periodontal disease, a well-known cause of tooth loss, causes the alveolar bone to dissolve [1,2,3,4]. Periodontal diseases are said to be one of the two most common diseases in dentistry, along with dental caries. Although periodontal diseases are caused by dental plaque or dental biofilm adhering to the border between the teeth and gingiva, its prevalence remains high worldwide, and it is the leading cause of tooth loss [5,6,7,8,9]. Periodontal diseases are characterized by progressive destruction of periodontal tissues caused by chronic inflammatory and immune responses induced by dental plaque, and recently, it has been strongly suggested to be closely related to various systemic conditions, including diabetes [10,11,12,13].

Recently, as the public’s knowledge of periodontal diseases has increased, an increasing number of patients are presenting to the hospital with symptoms of periodontal diseases as their main complaint. However, because periodontal diseases often progress slowly and do not cause pulpitis-like pain, they are easily overlooked, and detection is often delayed or treatment is neglected. Thus, understanding the actual condition of patients with periodontal diseases during clinic visits is very important for treatment [14,15,16,17].

The alveolar bone, similar to bones in other body parts, is repeatedly resorbed by osteoclasts and renewed by osteogenic cells [18,19,20,21,22,23,24,25]. In other words, an old bone is constantly being resorbed and replaced by a new bone. However, when there is an imbalance between the two, bone resorption progresses, and new bone formation cannot keep up. Periodontal diseases cause inflammation of periodontal tissues, and cytokines, which are substances produced during this inflammation process, promote the function of osteoclasts.

The intrinsic alveolar bone is the portion of the cortical bone that forms the inner wall of the alveolar bone, surrounds the tooth root, and contains Sharpey’s fibers, which have the important function of supporting the teeth. The supporting alveolar bone is the portion of the alveolar process other than the intrinsic alveolar bone that supports the intrinsic alveolar bone. Structurally, it is composed of cortical bone (the lateral surfaces of the buccal and lingual surfaces of the alveolar process) and trabecular bone. The cancellous bone has well-developed trabecular bone and bone marrow tissue, and when young, it is also involved in hematopoiesis.

This review describes the clinical practice, mainly surgical procedures, and the latest basic research on the alveolar bone, which plays an important role in supporting teeth and periodontal tissues.

### 1.1. Alveolar Bone Resorption

Alveolar bone resorption can be physiological or pathological, and the most common cause is chronic inflammation due to periodontal diseases [26,27,28]. Alveolar bone resorption can be divided into three types [29,30,31,32,33]:(1)Horizontal bone resorption. There is uniform horizontal resorption of the alveolar bone that occurs extensively on multiple teeth, e.g., the line connecting the cement–enamel border of the adjacent teeth is parallel to the alveolar crest.(2)Vertical resorption. There is vertical resorption of the alveolar bone from the apex, often confined to one or two teeth. The line connecting the cement–enamel border of the adjacent tooth and the alveolar crest is not parallel.(3)Mixed bone resorption. A combination of horizontal and vertical bone resorption occurs.

### 1.2. Alveolar Bone Surgery

Alveolar bone surgery is the general term for periodontal surgery aimed at removing gingival or periodontal pockets, and alveolar bone surgery is the main surgical technique used. Alveolar bone surgery includes alveolar osteoplasty, alveolar resection, and bone grafting, and it is often performed simultaneously with other periodontal surgical procedures.

Alveolar bone reshaping is a surgical procedure that corrects alveolar bone morphology to a physiological form without removing the native alveolar bone. Its indications include thick shelf-like bone margins, exostosis, and bony prominences. Generally, it is performed simultaneously with other periodontal surgical treatments.

### 1.3. Alveolar Osteotomy

In this surgical procedure, the alveolar bone is constructed closer to the physiological morphology by removing it, including the intrinsic alveolar bone [34,35]. The amount of alveolar bone to be removed must be determined so that the crown-to-root ratio does not worsen and the roots and root bifurcations are not exposed when the height of the alveolar bone is reduced. Indications include cratering bone defects in the interdental space, submarginal pockets, and crown lengthening. As with alveolar bone reshaping, this procedure is performed simultaneously with other periodontal surgical procedures [35,36,37,38,39].

## 2. Characteristics of Periodontal Treatment

Non-surgical treatments, including scaling and root planing, adjuvant treatments, including antibiotics and disinfectants, and lifestyle modifications, including smoking cessation and improvement of oral hygiene, play important roles in periodontal disease treatment.

Removing the cause of periodontitis improves or stops its progression. With basic periodontal treatment, mainly consisting of root cause elimination methods, mild periodontitis is restored to a healthy state, and progression is arrested [10,40,41]. However, moderate or severe periodontitis requires a more complex periodontal treatment with frequent use of periodontal surgery and restoration of oral function. It is difficult to achieve complete regeneration of lost periodontal tissue using current routine periodontal treatments, including periodontal tissue regeneration therapy.

As part of periodontal treatment, lifelong ongoing management using “Supportive Periodontal Therapy (SPT), maintenance, and treatment for the prevention of severe periodontal disease” are indispensable [42,43,44].

Periodontitis is highly likely to recur due to the constant presence of bacterial plaque and traumatic factors in the oral cavity, deep periodontal pockets and root bifurcation lesions that may persist despite appropriate periodontal treatment, and the influence of systemic factors over time. Therefore, when basic periodontal therapy, periodontal surgery, and oral function restoration therapy have “cured” or “stabilized” the disease, or when inflammation is present in the gingiva even though the probing depth is <4 mm, SPT, which is part of periodontal therapy, is used to maintain periodontal tissues for an extended period, while maintenance therapy, as a preventive therapy and health care, is used to prevent severe periodontal disease. SPT and periodontal disease severity prophylaxis consist mainly of plaque control, scaling and root planing, and occlusal adjustment by a dental healthcare professional. SPT and maintenance are essential because periodontal disease can easily recur with inadequate plaque control. Additionally, these treatments at appropriate intervals can preserve periodontal tissues, including the alveolar bone, allowing them to function for an extended period.

Although periodontal treatment is vital to health care, it also has disadvantages. Depending on the periodontal disease progression, pain may be experienced during treatment. When cleaning the grooves in the gums, which tend to be a breeding ground for periodontal disease bacteria, the instruments will touch the gums. For those with particularly advanced gum disease, the pain may also be more intense, and anesthesia may be necessary. Another disadvantage of periodontal treatment is that it requires more frequent and time-consuming visits. Periodontal disease treatment involves removing the tartar and plaque where the causative bacteria live. However, the tartar and plaque in the hard-to-see areas deep in the periodontal pockets may be missed. Consequently, due to care, such as brushing or lifestyle slackness, the remaining causative bacteria in the oral cavity may become active, causing periodontal disease recurrence. Therefore, early prevention must always be considered.

Periodontal patients with systemic or suspected systemic disease should be thoroughly interviewed medically before periodontal treatment starts, and promptly referred to a physician based on their symptoms [45,46,47]. Periodontal treatment may be possible with adequate management of systemic disease. However, depending on the systemic disease and its symptoms, periodontal and dental treatments may be difficult in some cases. Therefore, the patient should be referred early to a medical institution specializing in that systemic disease to provide the information necessary for the condition and treatment and to request management of the systemic disease [48,49].

### 2.1. New Classification of Periodontal Disease (2018 American Academy of Periodontology and European Federation of Periodontology)

Periodontal disease is a complex and multifactorial condition that requires a comprehensive approach to diagnose and treat. In November 2017, the American Academy of Periodontology (AAP) and the European Federation of Periodontology (EEP) co-sponsored the World Workshop on the Classification of Periodontal and Peri-Implant Diseases and Conditions.

The workshop results were summarized in a consensus report and published as a new classification of periodontal disease by the AAP/EFP during the EuroPerio9 in Amsterdam in June 2018. The most significant change in the new classification is that of periodontitis. The 1999 classification divided periodontitis into two major categories: invasive and chronic. However, the new classification combines these two categories into a single periodontitis and introduces a diagnostic framework of stages and grades. The severity and complexity of periodontitis are divided into four stages (stage I is the mildest and stage IV is the most severe), and the risk of periodontitis progression is divided into three grades (grade A is the lowest and grade C is the highest). Factors including smoking and diabetes are considered in determining the grade [50].

### 2.2. Tissue Engineering for Periodontal Tissue Regeneration

Adherence to the basic flow of periodontal treatment (i.e., examination and diagnosis, basic periodontal treatment, reevaluation, and maintenance or supportive periodontal therapy) can prevent the progression of periodontal diseases, and periodontal tissue healing can be promoted through the precise implementation of periodontal scaling and root planning and other root cause elimination therapies. Environmental and acquired factors that influence periodontal disease include stress stimuli. Stress stimuli are associated with the severity of periodontal disease, and the state of mental tension (stress response) induced by stress stimuli influences the immune response. European researchers have found a relationship between gingivitis and depression. They published a study showing that patients with gingivitis had 1.8 times higher rates of subsequent depression than those without gingivitis, highlighting the importance of maintaining good oral health [51]. This trend seems particularly high among young people. Immune inflammation is responsible for the correlation between gingivitis and depression, and it has been reported that pain, bad breath, prostheses, and poor oral hygiene can cause symptoms of depression. Along with personal, genetic, and systemic factors, co-factors including chronic stress and depression contribute to the development and progression of periodontal disease and peri-implantitis, compounding the resulting impact on treatment responsiveness [52]. The neurobiological and neurobehavioral pathogenic associations between chronic stress, depression, and systemic disease are mainly related to necrotic periodontal disease. In cases of periodontitis with severe alveolar bone resorption, periodontal pockets may remain even after a root cause elimination therapy; even if the periodontal pockets disappear, the healing process may be functionally and esthetically unacceptable, with significant gingival recession. In other words, periodontal tissue regeneration with a new alveolar bone and cementum formation cannot be expected, even with conventional cause elimination therapy and periodontal surgery [53,54,55] (Table 1 and Table 2). Therefore, there are great expectations for periodontal tissue regeneration therapy based on tissue engineering (Table 1 and Table 2).

Langer et al. proposed a new concept of tissue engineering [56]. This concept is based on the idea of constructing new tissues by combining the “cells” that make up the tissue, the “scaffold” that arranges the cells in three dimensions and gives the tissue its shape, and the “signaling molecules” that control cell functions, such as proliferation and differentiation [57,58,59,60,61,62,63]. This concept also promotes the development of regenerative medicine for various tissues and organs.

Regarding periodontal tissue regeneration, periodontal ligament-derived cells are selectively and preferentially induced on the root surface of periodontal tissue defects, undifferentiated mesenchymal cells contained in periodontal ligament-derived cells proliferate while retaining pluripotency and differentiate into hard tissue forming cells (i.e., osteoblasts and cementoblasts) and periodontal ligament fibroblasts in a site-specific manner, and periodontal ligament fibroblasts are differentiated from periodontal ligament cells in a site-specific manner [64,65,66,67,68]. To acquire new attachments, the collagen fiber bundles produced by periodontal ligament fibroblasts must be embedded in the new bone tissue and cementum created by osteoblasts and cementoblasts [69,70,71].

## 3. Guided Bone Regeneration (GBR)

GBR is a technique that applies the principles of guided tissue regeneration only to bone tissue. It is a method of bone regeneration by covering the bone defect with a membrane to prevent the invasion of epithelial cells and connective tissue-derived cells and to provide time and space for osteogenesis by osteoblasts. This is a technique used to achieve bone augmentation. GBR is also an integral component of today’s implant therapy. The membrane must have the following characteristics: biocompatibility, cell blocking, tissue integrity, space-making ability, and ease of use [72,73].

In several cases, autologous bone or various bone augmentation materials are grafted for space creation and membrane retention, and bone augmentation is performed using their osteoinductive or osteoconductive abilities.

### Clinical Applications of Nonabsorbable and Resorbable Bone-Replacing Materials

In several cases, autogenous bone grafting was required for space making, but a secondary wound had to be created for harvesting autogenous bone. The associated prolongation of operative time and the risk of complications, such as bleeding, swelling, pain, deformation of receptors, and nerve palsy, have made alternative materials desirable and increasingly used. Initially, hydroxyapatite (HA) was used as a substitute material for autologous bone [74].

Subsequently, various bone replacement materials, such as deproteinized bovine bone mineral and β-tricalcium phosphate (*β*-TCP) were developed and clinically applied. Among these, Bio-Oss, an inorganic bovine bone matrix, has become the most clinically applied bone replacement material in the world, partly because it coincides with the development of GBR.

Tricalcium phosphate (TCP) has been developed as a resorbable and bone replacement material for hyaluronan, which is basically non-resorbable. TCP is a material that is highly degradable and resorbable in body fluids compared with HA. *β*-TCP is stable at low temperatures, whereas α-TCP is stable at high temperatures [75]. Among these, *β*-TCP is currently the most widely used bone replacement material.

## 4. Clinical Applications of Membranes

Although the ePTEF membrane has been the gold standard in clinical applications since the mid-1990s, absorbable membranes consisting primarily of bovine or porcine tendon and skin-derived type I collagen have also been used [76,77]. In addition to conventional use, dPTFE membranes have been applied as open barrier membranes for ridge resurfacing and other applications [78]. The use of the GoreTex membrane was discontinued in 2012, after which the development and clinical application of different membranes accelerated.

In addition to dPTFE membranes, titanium mesh and titanium membranes are clinically applied as nonabsorbable membranes. Titanium micromesh was developed around 1990, and since then, various titanium meshes have been applied clinically. In the late 1990s, GBR using thin plate-shaped titanium as a barrier membrane was reported [79], and it has been improved and clinically applied since then. The absorbable membrane has been improved and applied clinically since then.

As absorbable membranes, both biologically derived collagen membranes and membranes prepared from artificial materials have been used in clinical applications. The membrane is made of two layers, and heparin sulfide and fibronectin are added to the inner layer to increase its ability to inhibit epithelial penetration toward the root apex [80]. The sausage technique reported by Urban et al. in 2013 is now widely applied clinically as a predictable bone augmentation method [81].

The goal of bone replacement materials is to achieve an osteogenic potential equivalent to that of autogenous bone with osteoinductive potential. Future prospects include the development of bone replacement materials that promote bone formation, such as composites of growth factors and stem cells in artificial bone with interconnecting holes, and membranes that have similar capabilities.

## 5. Implant Therapy

In implant therapy, the implant body is placed at the defective site in the jawbone [82,83,84,85]. A superstructure is attached to this artificial tooth root to restore masticatory function, pronunciation, and esthetics. Given their excellent biocompatibility, titanium and titanium alloys are the main materials used for implants, which are artificial tooth roots.

Natural teeth have cementum and periodontal ligaments, which are bound to the alveolar bone by connective tissue attachments, whereas the implant body is bound to the alveolar bone by osseointegration [86,87,88]. Therefore, in natural teeth, collagen fibers run perpendicular to the root and are embedded in the cementum, whereas in the peri-implant mucosa, they run parallel to the implant. In natural teeth, blood supply comes from the periodontal ligament, alveolar bone, and gingiva, whereas in implants, blood supply comes from the alveolar bone and gingiva. The spread of plaque-induced inflammation differs between periodontal and peri-implant tissues. In peri-implantitis, the inflammatory infiltrate can easily spread to the alveolar bone. Since periodontitis is considered a risk factor for peri-implantitis, implant therapy should be initiated after periodontal therapy [89,90,91]. In addition, implants are considered less resistant to inflammation than natural teeth; thus, regular management with attention to plaque control and occlusal control is important. Peri-implantitis is a bacterial infection; thus, eliminating inflammatory lesions in the affected area is prioritized when treating it [92,93,94]. It is necessary to simultaneously diagnose and treat periodontal disease in the affected area and in the remaining teeth. Whether conservative or surgical treatment is chosen, debridement of the bacterial biofilm attached to the microstructure of the implant surface is considered extremely important for successful treatment. First, treatment according to basic periodontal therapy is performed, and after the resolution of inflammation is observed, the patient is reevaluated, and the surgical procedure is chosen on the basis of the examination results. Periodontal histological examination of the remaining teeth and treatment of periodontal disease are essential.

## 6. Bone Grafting

Bone grafting is a periodontal tissue regeneration therapy, either alone or in combination with guided tissue regeneration (GTR) or enamel matrix derivatives [95,96,97,98]. Bone grafts are used to fill periodontitis-induced subgingival defects and improve bone morphology to halt the progression and prevent the recurrence of periodontitis. Compared with flap surgery, bone grafting may reduce probing depth, improve clinical attachment levels, and regenerate periodontal tissues (neointegration) with bone formation, cementum, and partially functional periodontal ligament formation in bone defects. Moreover, bone grafting is used for bone building before or concurrent with implant placement [99,100].

Bone grafting has osteogenic potential, osteoinductive potential, and osteoconductive potential. Osteogenesis is the formation of bone by preosteoblasts and mesenchymal cells in the graft material (autologous bone) [101,102,103]. The bone-inductive capacity is defined as bone formation by bone morphogenetic proteins (BMPs) in autologous or allogeneic bone [104,105,106]. The osteoinductive capacity is the ability of a bone graft material to promote continuous bone formation from existing bone by providing a scaffold for cells involved in bone formation and calcium and phosphorus for calcification.

### 6.1. Types and Characteristics of Bone Grafting Methods

Currently, four bone grafting methods for periodontal treatment use bone grafting materials: (1) autologous bone, (2) allogeneic bone (bone from other families), (3) heterologous bone, and (4) artificial bone.

(1)Autologous bone grafting

Autologous bone grafting is a transplantation method in which the bone is harvested from the patient’s body and transplanted into a bone defect [19,107,108,109]. The bone is harvested from both intraoral and extraoral sites. Intraoral sites for bone harvesting include the vicinity of the graft site, buccal shelf, edentulous jaw crest, exostoses, maxillary tuberosity, mandibular angle, mastoid, and extraction socket. Bone swaging is also used [110,111,112], in which a slit is made in the bone adjacent to the bone defect using a rotary cutting instrument to pull the bone into the defect. In contrast, extraoral bone is harvested from the bone marrow and iliac bone.

Autogenous bone has osteogenic and osteoinductive potential and is said to be free of immunogenic problems. Its advantages include complete resorption and replacement with a new bone through bone remodeling. However, in some cases, surgical invasiveness and restrictions on the site and amount of bone harvested make it difficult to obtain the necessary amount of bone for transplantation.

(2)Allogeneic bone grafting

Allografts and heterogeneous bone grafts include freeze-dried bone allografts, which use freeze-dried bone from a donor, and demineralized freeze-dried bone allografts [113,114,115,116].

(3)Bone xenografting

Bone xenografting is a bone grafting technique that uses non-human bone [117,118]. Currently, bovine bone is mainly used for periodontal treatment because it is similar in structure to human bone, biocompatible, and bioabsorbable [119,120,121,122]. Heterologous bone has an osteoconductive capacity and is readily available [123,124]. However, patients who are concerned about the risk of infection must be fully informed, and consent must be obtained before use.

(4)Artificial bone grafts

Synthetic grafting materials are used for artificial bone grafts [118,125,126]. These synthetic grafting materials include bioceramics and bioactive glass. Artificial materials are less surgically invasive, non-pathogenic, and non-infectious, with no limits on the supply volume. Bone formation and periodontal tissue regeneration are negligible, and they work primarily as filler spacers.

### 6.2. Indications

The indications are two- to three-walled, deep, and narrow bone defects in terms of defect site and morphology. The treatment results are also good. Stable wound healing and a secured space for tissue regeneration by sufficient coverage of the bone graft material with a gingival valve are important. A thin gingiva, gingival retraction, and cratering of the gingiva will negatively affect the treatment outcomes.

### 6.3. Key Points of the Procedure

(1)For preoperative preparation, thorough plaque control by the patient is important [127,128,129]. In the case of upset teeth, wound healing stability may be impaired, and the prognosis may be poor; thus, occlusal adjustment and splint therapy should be used to reduce excessive tooth movement and fremitus.(2)In bone grafting, the incision line and gingival valve should be preserved to ensure that the bone graft material is completely covered by the gingiva and retained within the bone defect [95,130]. Incisions close to or over the bone defect should be avoided. Soft tissue involvement should be carefully minimized during gingival valve dissection.(3)For receptor preparation, the root surface and bone defect, which is the graft bed, should be debrided by hand or by a rotary cutting instrument, such as an ultrasonic scaler or finishing bar. The cortical bone may be perforated using a round bar to supply blood to the defect.(4)Preparation of bone graft materials

For autogenous bone [131,132,133], cortical bone fragments are harvested from around the surgical site using bone chisels or bone forceps, and bone aggregates are harvested using rotary or ultrasonic cutting instruments. A bone blend of cortical and cancellous bone is collected using a trephine bar. The bone graft material should not be in contact with the saliva during collection and should not be dried until transplantation.

(5)Filling and suturing bone graft materials

Before use, the bone graft material should be infiltrated with saline solution, excess water should be removed using gauze, the material should be brought into the bone defect using an amalgam carrier or spatula, and the defect should be filled with a small amount of bone graft material. At this time, a space is left for osteoblasts and mesenchymal cells to infiltrate between bone graft materials. To achieve primary wound healing, the wound should be sutured to ensure that a tension-relieving incision and secure gingival valve reinstatement are possible, if necessary. The gingival valve should completely cover the bone graft material and avoid excessive tension. Simple or mattress sutures with monofilament or nylon thread should be used to avoid breaking the filled bone graft material. Postoperative management, such as instructing the patient not to brush the affected area for the first week after surgery, is essential.

## 7. New Findings from Clinical and Basic Research on the Alveolar Bone

Clinical and basic research on the alveolar bone is ongoing worldwide, and new findings are reported continuously. This section discusses the most recent findings.

Periodontitis is a major cause of tooth loss, but no effective treatment has been established to restore the inflammatory bone lost because of periodontitis. Exosomes are essential for tissue regeneration, as paracrine factors of mesenchymal stem cells (MSCs) [134,135,136]. Lei et al. investigated whether exosomes secreted by periodontal ligament stem cells (PDLSCs) could be therapeutic agents against bone loss in periodontitis [137]. Exosomes secreted from healthy periodontal ligament-derived PDLSCs (h-PDLSCs) and their functions were evaluated against PDLSCs isolated from the inflammatory periodontium of patients with periodontitis (i-PDLSCs) [137]. Mechanistically, h-PDLSC exosomes suppress the hyperactivation of canonical Wnt signaling and restore the osteogenic differentiation potential of inflammatory PDLSCs [137]. These findings suggest that MSC-derived exosomes are an effective and practical cell-free MSC therapy for periodontitis.

Ward focused on tissue engineering and periodontal tissue regeneration [55]. Tissue engineering has the potential to restore the normal physiological function and health of diseased sites. This paper focuses on the application of tissue engineering to periodontal diseases in canines and other animal species, with potential applications in veterinary medicine [55]. Key areas of tissue engineering include scaffolds, signaling molecules, stem cells, and gene therapy. To date, the results are still unpredictable [55]. However, tissue engineering has proven successful in regenerating lost periodontal tissue, and new possibilities for treating this veterinary case are discussed.

Al-Sosowa et al. investigated alveolar bone density and thickness in Chinese participants with and without periodontitis [138]. In a retrospective and cross-sectional study, alveolar bone volume, bone density, and bone thickness in approximately 668 mandibular molars (344 periodontally healthy teeth and 324 periodontally inflamed teeth) were evaluated using cone-beam computed tomography [138]. Comparative statistical tests were performed for age, sex, tooth type, tooth side, and degree of bone loss. The results revealed that alveolar bone thickness and density decreased in teeth affected by periodontal diseases, indicating that periodontal disease is still a factor that affects the alveolar bone [138].

Takedachi et al. evaluated the safety and efficacy of periodontal tissue regeneration using adipose tissue-derived multilineage progenitor cell (ADMPC) autologous transplantation [139]. Twelve patients with periodontitis enrolled in an open-label, single-arm, exploratory phase I clinical trial in which ADMPC isolated from subcutaneous adipose tissues were autologously transplanted [139]. Each patient underwent flap surgery, at which time the autologous ADMPC was transplanted into the bone defect along with a fibrin carrier. The results suggest that ADMPC autografting may be a treatment option for severe periodontitis by inducing periodontal regeneration [139]. We predict that ADMPC autografts have great potential as a new treatment for alveolar bone regeneration and will continue to attract attention.

Bone MSC (BMSC) transplantation promotes bone repair and regeneration. BMSCs die within a short period after transplantation, and they generate large amounts of apoptotic cell-derived extracellular vesicles (Apo-EVs) during apoptosis in the craniofacial bone [140]. Li et al. evaluated the in vivo osteogenic potential of Apo-EVs in a rat critical-size bone defect model to clarify the potential role of Apo-EVs in the repair and regeneration of craniofacial bone defects [140]. They found that local transplantation of Apo-EVs enhanced bone formation [140]. Apo-EVs promote bone regeneration of parietal bone defects and may be a promising strategy for craniofacial bone repair and regeneration because they avoid the side effects and limitations of BMSC therapy.

Phosphoinositide 3-kinase (PI3K) is found intracellularly and is involved in the regulation of cell survival, proliferation, apoptosis, and angiogenesis [141,142,143]. Wang et al. investigated the role of PI3K in the bone-destructive process of periodontitis and provided data that could be helpful in its treatment [144]. Relative mRNA expression levels of PI3K, Acp5, and NFATc1 in the normal human periodontal ligament and chronic periodontitis were analyzed by quantitative reverse-transcription polymerase chain reaction [144]. In a mouse model of apical periodontitis with root canals exposed in the oral cavity and an inflammatory environment established at the osteoclast and osteoblast levels, the expression of PI3K, Acp5, and NFATc1 genes in the chronic apical periodontitis group was significantly increased (*p* < 0.05) compared with that in the healthy apical tissue group [144]. In the *Escherichia coli* lipopolysaccharide (LPS)-mediated inflammatory microenvironment, the gene and protein expressions of PI3K, TRAP, and NFATc1 in osteoclasts were significantly increased compared with those of normal controls (*p* < 0.05). In contrast, the gene and protein expressions of PI3K, BMP−2, and Runx2 in osteoblasts were significantly decreased in the inflammatory microenvironment (*p* < 0.05) [144].

Marginal zone B and B−1 cell-specific protein (MZB1), a novel molecule associated with periodontitis, is an endoplasmic reticulum-localized protein, and its increased expression has been associated with various human diseases [145,146,147]. However, a few studies have examined the effect of MZB1 on human periodontal ligament cells (hPDLCs) in the presence of periodontitis and the mechanisms underlying this effect. Li et al. performed experiments to determine the role of hPDLCs in migration and alveolar bone organization [148]. Bioinformatics analysis revealed that MZB1 is one of the most significantly upregulated genes in patients with periodontitis. mZB1, as a target gene of miR-185-5p, plays an important role in suppressing hPDLC migration through the NF-κB signaling pathway and exacerbating alveolar bone loss [148]. Bioinformatics analysis was used in this study [148]. This analytical method has attracted much attention in oral research, and with this analysis, various mechanisms of alveolar bone will be revealed in the future.

Hou et al. explored the effect of A20 on macrophage polarization in periodontitis and its underlying mechanisms [149]. A20 has been reported to be involved in inflammation and bone metabolism in periodontitis, and the regulation of macrophage polarization may be an effective target for periodontitis treatment [150,151,152]. They used an adeno-associated virus targeting A20 to knockdown or overexpress A20 in the periodontal tissues of mice with experimental periodontitis [149]. A20 knockdown increased the expression of the NLRP3 inflammasome pathway in mouse periodontal tissues and THP−1 cells [149]. Conversely, A20 overexpression inhibited the NLRP3 inflammasome pathway. MCC950 suppressed M1 macrophage polarity, which was exacerbated by A20 knockdown in LPS- and IFN-γ-stimulated cells [149]. This finding suggests that A20 inhibits periodontal bone resorption and NLRP3-mediated M1 macrophage polarization and may be a novel target for the treatment of periodontitis.

Leptin is secreted by adipocytes and strongly inhibits feeding and increased energy expenditure mainly through receptors in the hypothalamus, and its inaction is thought to be important in the etiology of obesity [153,154,155,156,157]. Despite accumulating evidence that leptin plays an important role in periodontal diseases, the mechanism by which it promotes periodontitis development is unknown. Han et al. found elevated leptin expression in periodontitis mouse serum compared with healthy controls, suggesting that leptin exacerbates the ligature response of periodontal diseases by promoting polarization of M1 macrophages via the NLRP3 inflammasome from the Nlrp3-/- periodontitis model [158]. Moreover, they reported that leptin promoted the progression of periodontitis via the polarization of inflammatory M1 macrophages. Periodontitis is characterized by alveolar bone resorption [158]. In the future, targeting leptin/NLRP3 signaling may be applied to elucidate the mechanisms of the alveolar bone and periodontal diseases. Furthermore, we hope that it will be applied to alveolar bone treatment.

*Porphyromonas gingivalis*-derived LPS (Pg-LPS) promotes inflammation, osteoclastogenesis, and alveolar bone resorption and is considered a major pathogenic factor in chronic periodontitis [159,160,161,162]. The thrombomodulin (TM) lectin domain (TMD1) exerts an anti-inflammatory function. Thus, Chang et al. examined the therapeutic effect of recombinant TMD1 (rTMD1), which inhibits Pg-LPS-induced osteoclastogenesis and alveolar bone loss [163]. They reported that rTMD1 inhibits Pg-LPS-enhanced M1 macrophage polarization, osteoclastogenesis, and periodontal bone resorption [163]. Since osteoclastogenesis is a very important factor for the alveolar bone, rTMD1 will continue to be of interest.

Induced pluripotent stem (iPS) cells are pluripotent stem cells that can be produced via the introduction of a few genes into somatic cells. The technology for regenerating periodontal tissues, including alveolar bone, is expected to be applied in the treatment of diseases, such as tooth loss and periodontitis [164,165,166,167,168]. Kawai et al. established a method for inducing bone differentiation from iPS cells to form bone-like nodules in a brief period [169]. The induced bonelike nodules contain osteoblast-like and osteocyte-like cells, and their process of differentiation is similar to the process of in vivo differentiation, which Kawai et al. reported could be applied to research on the differentiation process of osteogenic cells [169]. It has been reported that bone formation ability (e.g., calcification) increases when mesenchymal stem cells (MSCs) derived from iPS cells are cultured with calcium phosphate. Thus, iPS cell-derived MSCs are expected to be applied in dental regenerative medicine.

The newest research in the regenerative approach to alveolar bone regeneration with 3D printed and colonized scaffolds. For example, research has been conducted on the fabrication and in vitro characterization of novel 3D-printed hydroxyapatite scaffolds. Various proof-of-concept studies have been published on the biocolonization of 3D-printed hydroxyapatite scaffolds using mesenchymal stem cells [170].

Regenerative medicine has recently been attracting attention. Methods for the creation of induced pluripotent stem cells have been developed, and research on regenerative medicine in various organs is becoming more active. Moreover, currently, bone and cartilage regeneration, including dentistry, has most often been used in regenerative medicine in clinical practice. Therefore, clinical and basic research on alveolar bone mechanisms, especially alveolar bone regeneration, will become increasingly active in the future.

## 8. Conclusions

This review acknowledges the limitation that only the MEDLINE database of PubMed provides other sources of information, such as gray literature, conference proceedings, and clinical practice guidelines. One limitation is the lack of previous research in our field. Writing a literature review is an important task for scientific research because it helps us determine the scope of the current research in our chosen field. Literature reviews provide a source of information for researchers who need scholarly information to achieve a specific goal or objective; however, when the focus is on a current and developing alveolar research problem or a very narrow research question, it is possible that limited or no previous research on the topic can be found and presented.

This review describes clinical practice, mainly surgical procedures, and the latest basic research on the alveolar bone, which plays an important role in supporting teeth and periodontal tissues. The oral cavity is the gateway to the body and shows a significant association with various diseases. Similarly, the mechanism of the alveolar bone will continue to be elucidated in more detail. Alveolar bone treatment methods, as described herein, are advancing annually, and regenerative therapies will be the mainstay of treatment in the future. The relationship among oral diseases, oral bacteria, systemic diseases, and alveolar bone is currently the subject of epidemiological studies; thus, elucidating the detailed mechanisms is necessary. The acquisition and maintenance of a healthy oral environment and alveolar bone will have both direct and indirect positive effects on overall health.

## Figures and Tables

**Figure 1 biomedicines-11-00843-f001:**
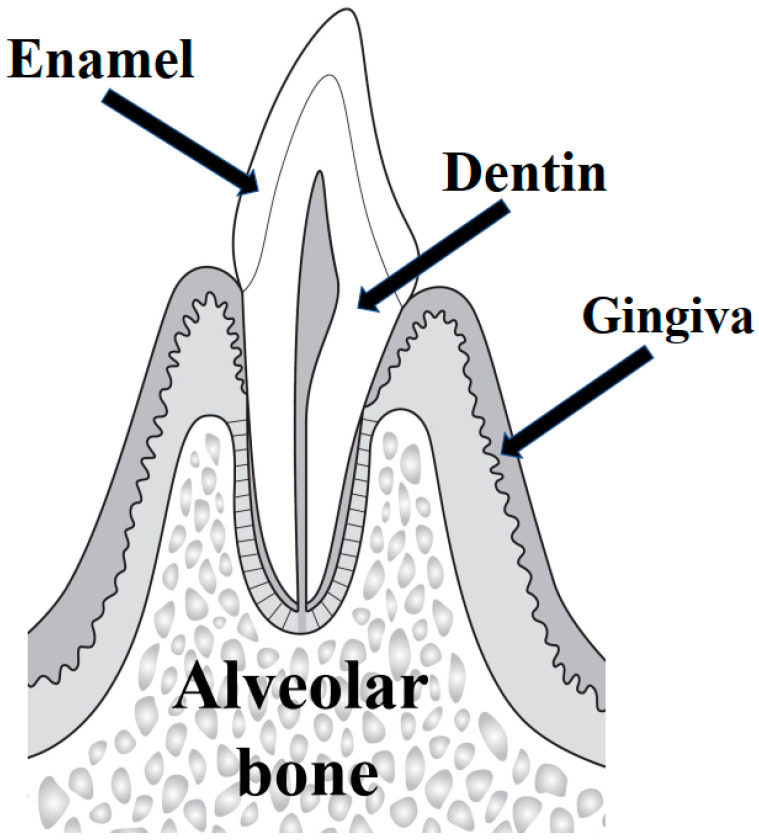
The alveolar bone. A diagram of a tooth cut vertically is shown. The alveolar bone is the bone that supports the tooth and is connected to the root of the tooth via the periodontal ligament, which receives the force of the bite. The outer surface is made of hard “dense bone”, while the inner surface is made of soft “trabecular bone”, and this outer frame and cushioning make it resistant to impact.

**Table 1 biomedicines-11-00843-t001:** Types of Alveolar Bone Regeneration Treatment.

Type	Summary	Application	Characteristics
**Sinus lift**	Sinus lift is a technique used to place implants in the back teeth of the upper jaw after the bone thickness has been supplemented.	Sinus lift is an alveolar bone regeneration technique performed when the maxillary bone is not thick enough. This technique is performed to build a bone by filling the space above the alveolar bone, called the maxillary sinus, with a bone filler or other materials.	**Advantages:** Sinus lift is a treatment method that allows extensive bone creation so that longer implants can be used. It is a treatment method known worldwide for its long-lasting effects, and many articles have been written about it. Additionally, because the treated area is visible, the risk is relatively low. **Disadvantages:** Extensive osteogenesis means a larger surgical area, and the treatment period is longer if the sinus lift and implant placement are done separately as a two-stage procedure. The use of expensive bone replacement materials also places a burden on the patient’s body and finances.
**Socket lift**	Socket lift is a type of osteogenic surgery applied to maxillary implant treatment. The bottom of the maxillary sinus is surgically lifted to create a space, and a bone replacement material (either your own bone or artificial bone) is grafted into the space to regenerate the alveolar bone (the bone that supports the teeth). When an implant is placed in the upper jaw, if the alveolar bone is still thin, it is impossible to place and settle the implant because of the weak foundation. In such cases, the socket lift regenerates the alveolar bone through regenerative surgery, allowing the implant body to be placed in the maxillary region.	Like sinus lift, socket lift is an alveolar bone regeneration technique performed when the maxillary bone is not thick enough. This technique is performed to build a bone by filling the space above the alveolar bone, called the maxillary sinus, with a bone filler and other materials. It is performed when the thickness of the maxillary bone is generally 5 mm or more.	**Advantages:** It is a type of bone regeneration surgery that involves the grafting of bone augmentation material, but the scope of the surgical procedure can be reduced. The surgical procedure is less stressful on the body. The procedures for placing the implant body can be performed at the same time. Therefore, the overall treatment time can be shortened. **Disadvantages:** It is assumed that a certain amount of alveolar bone thickness (3–5 mm) remains. If the alveolar bone is not more than 5 mm, the treatment may not be possible. Furthermore, the scope of the surgical procedure is narrow. It can reduce the burden on the body, but the range of bone that can be regenerated is also narrower. The condition of the maxillary sinus cannot be visually confirmed during the surgery. Therefore, there is a risk of damage to the maxillary sinus, resulting in sinusitis or sinusitis.
**GBR (guided bone regeneration)**	GBR is a method of regenerating the alveolar bone when the bone thickness or height is insufficient for implant treatment. Once the alveolar bone is lost due to bone resorption caused by severe periodontal disease or bone loss after tooth extraction, it becomes difficult to place an implant in that area. In such bone-deficient areas, “fibroblasts”, which do not form bone, tend to proliferate more easily than “osteoblasts”, which form bone. Therefore, in GBR, the area to be treated is covered with a “membrane” to prevent fibroblasts from invading the area and filling it with autologous bone or an artificial bone replacement material to promote the proliferation of osteoblasts.	While sinus lift and socket lift are performed to build bone inside the alveolar bone (jaw bone), GBR is performed to build bone outside the alveolar bone (i.e., the surface where the bone meets the gums). It is performed when the alveolar bone is not wide or high enough.	**Advantages:** By increasing the amount of bone, it is possible to “secure the amount of bone” necessary for implant placement and to place the implant in the appropriate position, which results in “high stability”. **Disadvantages:** When using autologous bone, “bone harvesting surgery” is required separately from implant placement, “treatment time is long”, and “it is not suitable for smokers and diabetics”.
**Socket preservation**	Socket preservation is an alveolar bone preservation technique used to minimize bone resorption by placing a bone filler in the extraction socket and sealing the gingiva at the time of tooth extraction. This technique is effective in cases of significant alveolar bone resorption or when a tooth is to be extracted, but an implant is desired in the future.	Socket preservation is a prophylactic technique to prevent “bone resorption” after tooth extraction. This technique is performed in conjunction with tooth extraction.	**Advantages:** The purpose of extraction is to prevent bone resorption after tooth extraction and to preserve the alveolar crest morphology rather than to perform an osteogenic procedure. If the alveolar crest is preserved, it can be used to support later dental implants or bridges. **Disadvantages:** This technique is not applicable if bone resorption is severe during tooth extraction.

**Table 2 biomedicines-11-00843-t002:** Periodontal Tissue Regeneration Materials Used in Periodontal Tissue Regeneration Therapy.

Product Name	Summary	Characteristics
**REGROTH^®^** **(Active ingredient: Trafermin (genetical recombination), 0.3% basic Fibroblast Growth Factor: FGF-2, Kaken Pharmaceutical Co., LTD., Tokyo, Japan)**	An artificially purified basic fibroblast growth factor, which is expected to regenerate lost alveolar bone and periodontal ligament by increasing the proliferation of periodontal tissue cells.	It can regenerate bone, cementum, and periodontal ligament of a tooth that is about to fall out. It has a high ability to regenerate the alveolar bone.
**Emdogain^®^ Gel** **(Straumann, Basel, Switzerland)**	A periodontal tissue regeneration material containing enamel matrix protein, which plays a vital role in tooth development. This protein may be useful for periodontal tissue regeneration.	Because of the large number of cases treated, there is a wealth of evidence (e.g., papers) and a relatively high degree of certainty. It is compatible with various bone replacement materials used as bone substitutes and can be easily used in combination with bone replacement materials.
**Geistlich Bio-Oss^®^ (Geistlich Pharma AG, Wolhusen, Switzerland)**	A bone grafting material, which is expected to promote bone regeneration when applied to bone defects caused by periodontal disease.	It is a natural bovine-derived porous bone filler with excellent osteoconductivity. It promotes growth and provides the bone volume necessary for implant placement in alveolar bone regeneration and augmentation. It is also highly compatible with the human body. It reduces the risk of infection.
**Cytrans^®^ Granules (Carbonated apatite granules, GC Corporation, Tokyo, Japan).**	A bone grafting material composed primarily of carbonate apatite, which is expected to promote bone regeneration when applied to bone defects caused by periodontal disease.	The main component, carbonate apatite, has the same composition as the bone, so it is efficiently replaced by the patient’s own bone, achieving strong osseointegration while maintaining the target bone surface height.
**Cytrans^®^ Elashield (GC Corporation, Tokyo, Japan).**	An absorbable membrane prepared from a chemically synthesized heavy polymeric material, which is expected to promote bone regeneration when applied to alveolar bone defects along with bone grafting materials.	The raw material is a fully synthetic polymer with a long absorption period suitable for GBR. It is designed to be easy to use and prevent postoperative gingival fissures because of its moderate firmness, flexibility, and ability to follow the shape of the bone. It can be used not only with autologous bone but also with bone replacement materials such as Cystrans Granules.
**Geistlich Bio-Gide^®^ (Geistlich Pharma AG, Wolhusen, Switzerland)**	An absorbable collagen membrane, which is expected to promote bone regeneration when applied to alveolar bone defects along with bone graft materials.	It has excellent tissue integration and strongly promotes bone regeneration. Due to its natural collagen structure, it is less prone to rupture than other membranes, resulting in fewer problems. The procedure is simple because membrane removal is not required. The burden on the patient is greatly reduced.

## Data Availability

Data sharing is not applicable to this article as no new data were created or analyzed in this study.

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
