# Peer review of "Recent Clinical Treatment and Basic Research on the Alveolar Bone"

_biomedicines, 2023, doi:10.3390/biomedicines11030843_

Round 1
Reviewer 1 Report
Please specify the limitations of the review.
How were the articles selected?
Is the review registered in prospero?
Author Response
Reviewer 1
Please specify the limitations of the review.
Response
We thank you for this crucial comment. We have revised the following (Conclusion section) “This review acknowledges the limitation that only the MEDLINE database of PubMed other sources of information such as gray literature, conference proceedings, and clinical practice guidelines.”
How were the articles selected?
Response
Thank you for pointing this out. We based this review on research articles published within the last five years.
Is the review registered in prospero?
Response
We thank you for this crucial comment. We have not yet registered in prospero, but will consider doing so. We thank you for considering and committing to reading our manuscript in detail and for making this crucial comment. We have revised the manuscript with new information and other changes in line with the reviewers’ comments.
The additions and corrections are indicated in red. We appreciate your review.

Reviewer 2 Report
Dear Authors,
the article is well written and organized.
The article needs few adjustments:
Check well the spelling (for example line 61 “func-tion”).
Line 106: I suggest to add some details about the treatment of periodontal disease and to mention the correlation between periodontitis, stress and depression.
I suggest you to add this reference: D’Ambrosio, F.; Caggiano, M.; Schiavo, L.; Savarese, G.; Carpinelli, L.; Amato, A.; Iandolo, A. Chronic Stress and Depression in Periodontitis and Peri-Implantitis: A Narrative Review on Neurobiological, Neurobehavioral and Immune–Microbiome Interplays and Clinical Management Implications. Dent. J. 2022, 10, 49. https://doi.org/10.3390/dj10030049
Line 198: I suggest to add some details about therapy of peri-implantitis.
Author Response
Reviewer 2
the article is well written and organized.
The article needs few adjustments:
Response
We thank you for this crucial comment. Our changes are indicated in our point-by-point responses given below to the comments. All revisions in the text are indicated with red letters.
- Check well the spelling (for example line 61 “func-tion”).
Response
We thank you for this crucial comment. We have checked and revised the spelling again.
- Line 106: I suggest to add some details about the treatment of periodontal disease and to mention the correlation between periodontitis, stress and depression.
Response
We thank you for this crucial comment. We have added the treatment of periodontal disease to the 107 line of page 3 (Characteristics of Periodontal Treatment section).
We added, “Environmental and acquired factors that influence periodontal disease include stress stimuli. Stress stimuli are associated with the severity of periodontal disease, and the state of mental tension (stress response) induced by stress stimuli influences the immune response. European researchers found a relationship between gingivitis and depression. They published a study showing that patients with gingivitis had 1.8 times higher rates of subsequent depression than those without gingivitis, highlighting the importance of maintaining good oral health. This trend seems particularly high among young people. Immune inflammation is responsible for the correlation between gingivitis and depression, and it has been reported that pain, bad breath, prostheses, and poor oral hygiene, can cause symptoms of depression.”
- I suggest you to add this reference: D’Ambrosio, F.; Caggiano, M.; Schiavo, L.; Savarese, G.; Carpinelli, L.; Amato, A.; Iandolo, A. Chronic Stress and Depression in Periodontitis and Peri-Implantitis: A Narrative Review on Neurobiological, Neurobehavioral and Immune–Microbiome Interplays and Clinical Management Implications. Dent. J. 2022, 10, 49. https://doi.org/10.3390/dj10030049
Response
We have added the following: ”Along with personal, genetic, and systemic factors, co-factors, including chronic stress and depression, contribute to the development and progression of periodontal disease and peri-implantitis, compounding the resulting impact on treatment responsiveness. The neurobiological and neurobehavioral pathogenic associations between chronic stress, depression, and systemic disease are mainly related to necrotic periodontal disease.”
- Line 198: I suggest to add some details about therapy of peri-implantitis.
Response
Thank you very much for your valuable comments on our manuscript. We have added, “Peri-implantitis is a bacterial infection; thus, eliminating the inflammatory lesions in the affected area is prioritized when treating it. It is necessary to simultaneously diagnose and treat the periodontal disease in the affected area and in the remaining teeth. Whether conservative or surgical treatment is chosen, debridement of the bacterial biofilm attached to the microstructure of the implant surface is considered extremely important for successful treatment. First, treatment according to basic periodontal therapy is performed, and after the resolution of inflammation is observed, the patient is reevaluated, and the surgical procedure is chosen on the basis of the examination results. Periodontal histological examination of the remaining teeth and treatment of the periodontal disease is essential.”

Reviewer 3 Report
This manuscript is a review that discusses the periodontal ligament, the effects of periodontal disease on the alveolar bone, and recent advances in treatment modalities, including implant therapy and induced pluripotent stem cell technology for periodontal tissue regeneration. It also describes clinical practices, techniques, and the latest basic research on the subject.
I would like to point out some limitations of this review and suggest possible improvements.
· - First, the review seems to have a narrow focus on alveolar bone and its regeneration, without considering the broader context of periodontal disease and its treatment. This could limit the relevance of the review to a wider audience, as periodontal disease is a complex and multifactorial condition that requires a comprehensive approach to diagnosis and treatment. The review could be improved by providing a more comprehensive overview of the current state of knowledge of periodontal disease, including its etiology, diagnosis and treatment.
· Second, the review appears to rely heavily on surgical interventions to treat periodontal disease, without considering the potential role of non-surgical interventions such as scaling and root planning, adjunctive therapies (e.g., antibiotics, antiseptics) and lifestyle changes (e.g., smoking cessation, improved oral hygiene). The review could be improved by providing a more balanced overview of the different treatment options for periodontal disease and their respective merits and limitations.
· Third, the review acknowledges the limitation that only the Medline database of PubMed was used, but it does not justify why this database was selected or whether other databases were searched. The review could be improved by a more comprehensive search strategy that includes multiple databases and other sources of information such as grey literature, conference proceedings and clinical practice guidelines.
· It also does not reference the newest research in regenerative approach to alveolar bone regeneration with 3D printed and colonized scaffolds. For example a research on Fabrication and in vitro characterization of novel 3D printed hydroxyapatite scaffolds. There are various proof-of-concept studies published on the biocolonization of 3D-printed hydroxyapatite scaffolds with mesenchymal stem cells. For example, https://doi.org/10.3390/ijms232314870
In summary, this review has several imperfections related to its narrow focus, limited consideration of non-surgical interventions and incomplete search strategy. These limitations can be addressed by providing a more comprehensive overview of periodontal disease, considering a broader range of treatment options, and expanding the search strategy to include multiple databases and other sources of information.
Author Response
Reviewer 3
This manuscript is a review that discusses the periodontal ligament, the effects of periodontal disease on the alveolar bone, and recent advances in treatment modalities, including implant therapy and induced pluripotent stem cell technology for periodontal tissue regeneration. It also describes clinical practices, techniques, and the latest basic research on the subject.
I would like to point out some limitations of this review and suggest possible improvements.
Response
Thank you very much for your valuable comments on our manuscript. We have revised our manuscript according to your comments and suggestions.
1 First, the review seems to have a narrow focus on alveolar bone and its regeneration, without considering the broader context of periodontal disease and its treatment. This could limit the relevance of the review to a wider audience, as periodontal disease is a complex and multifactorial condition that requires a comprehensive approach to diagnosis and treatment. The review could be improved by providing a more comprehensive overview of the current state of knowledge of periodontal disease, including its etiology, diagnosis and treatment.
Response
We thank you for this crucial comment. We have added the following (new section): “New Classification of Periodontal Disease (2018 American Academy of Periodontology and European Federation of Periodontology): Periodontal disease is a complex and multifactorial condition that requires a comprehensive approach to diagnose and treat. In November 2017, the American Academy of Periodontology (AAP) and the European Federation of Periodontology (EEP) co-sponsored the World Workshop on the Classification of Periodontal and Peri-Implant Diseases and Conditions.
The workshop results were summarized in a consensus report and published as the new classification of periodontal disease by the AAP/EFP during the EuroPerio9 in Amsterdam in June 2018. The most significant change in the new classification is that of periodontitis. The 1999 classification divided periodontitis into two major categories: invasive and chronic. However, the new classification combines these two categories into a single periodontitis and introduces a diagnostic framework of stages and grades. The severity and complexity of periodontitis are divided into four stages (stage I is the mildest and stage IV is the most severe) and the risk of periodontitis progression into three grades (grade A is the lowest and grade C is the highest). Factors, including smoking and diabetes, are considered in determining the grade.”
- Second, the review appears to rely heavily on surgical interventions to treat periodontal disease, without considering the potential role of non-surgical interventions such as scaling and root planning, adjunctive therapies (e.g., antibiotics, antiseptics) and lifestyle changes (e.g., smoking cessation, improved oral hygiene). The review could be improved by providing a more balanced overview of the different treatment options for periodontal disease and their respective merits and limitations.
Response
We thank you for this crucial comment. We have added the following (new section) “
Characteristics of Periodontal Treatment
Non-surgical treatments, including scaling and root planing, adjuvant treatments, including antibiotics and disinfectants, and lifestyle modifications, including smoking cessation and improvement of oral hygiene, play important roles in periodontal disease treatment.
Removing the cause of periodontitis improves or stops its progression. With basic periodontal treatment, mainly consisting of root cause elimination methods, mild periodontitis is restored to a healthy state, and progression is arrested. However, moderate or severe periodontitis requires a more complex periodontal treatment with frequent use of periodontal surgery and restoration of oral function. It is difficult to expect complete regeneration of lost periodontal tissue using current routine periodontal treatments, including periodontal tissue regeneration therapy.
As a part of periodontal treatment, lifelong ongoing management “Supportive Periodontal Therapy (SPT), maintenance, and treatment for the prevention of severe periodontal disease” are indispensable.
Periodontitis is highly likely to recur due to the constant presence of bacterial plaque and traumatic factors in the oral cavity, deep periodontal pockets and root bifurcation lesions that may persist despite appropriate periodontal treatment, and the influence of systemic factors over time. Therefore, when basic periodontal therapy, periodontal surgery, and oral function restoration therapy have “cured” or “stabilized” the disease, or when inflammation is present in the gingiva even though the probing depth is < 4 mm, SPT, which is part of periodontal therapy, is used to maintain periodontal tissues for an extended period, while maintenance therapy as a preventive therapy and health care is used to prevent severe periodontal disease. SPT and periodontal disease severity prophylaxis consist mainly of plaque control, scaling and root planing, and occlusal adjustment by a dental healthcare professional. SPT and maintenance are essential because periodontal disease can easily recur with inadequate plaque control. Additionally, these treatments at appropriate intervals can preserve periodontal tissues, including the alveolar bone, allowing them to function for an extended period.
Although periodontal treatment is vital to health care, it also has disadvantages. Depending on the periodontal disease progression, pain may be experienced during treatment. When cleaning the grooves in the gums, which tend to be a breeding ground for periodontal disease bacteria, the instruments will touch the gums. For those with particularly advanced gum disease, the pain may also be more intense, and anesthesia may be necessary. Another disadvantage of periodontal treatment is that it requires more frequent and time-consuming visits. Periodontal disease treatment involves removing the tartar and plaque where the causative bacteria live. However, the tartar and plaque in hard-to-see areas deep in the periodontal pockets may be missed. Consequently, due to care such as brushing or lifestyle slackness, the remaining causative bacteria in the oral cavity may become active, causing periodontal disease recurrence. Therefore, early prevention must always be considered.
Periodontal patients with systemic or suspected systemic disease should be thoroughly interviewed medically before periodontal treatment starts and promptly referred to a physician based on their symptoms. Periodontal treatment may be possible with adequate management of the systemic disease. However, depending on the systemic disease and its symptoms, periodontal and dental treatments may be difficult in some cases. Therefore, the patient should be referred early to a medical institution specializing in that systemic disease to provide information necessary for the condition and treatment and to request management of the systemic disease.”
- Third, the review acknowledges the limitation that only the Medline database of PubMed was used, but it does not justify why this database was selected or whether other databases were searched. The review could be improved by a more comprehensive search strategy that includes multiple databases and other sources of information such as grey literature, conference proceedings and clinical practice guidelines.
Response
We thank you for this crucial comment. We have revised to the following (Conclusion section) “This review acknowledges the limitation that only the MEDLINE database of PubMed other sources of information such as gray literature, conference proceedings, and clinical practice guidelines.”
- It also does not reference the newest research in regenerative approach to alveolar bone regeneration with 3D printed and colonized scaffolds. For example a research on Fabrication and in vitro characterization of novel 3D printed hydroxyapatite scaffolds. There are various proof-of-concept studies published on the biocolonization of 3D-printed hydroxyapatite scaffolds with mesenchymal stem cells. For example, https://doi.org/10.3390/ijms232314870
Response
We have added (New findings from clinical and basic research on the alveolar bone section)“The newest research in the regenerative approach to alveolar bone regeneration with 3D printed and colonized scaffolds. For example, a research on the fabrication and in vitro characterization of novel 3D-printed hydroxyapatite scaffolds. There are various proof-of-concept studies published on the biocolonization of 3D-printed hydroxyapatite scaffolds using mesenchymal stem cells.”
In summary, this review has several imperfections related to its narrow focus, limited consideration of non-surgical interventions and incomplete search strategy. These limitations can be addressed by providing a more comprehensive overview of periodontal disease, considering a broader range of treatment options, and expanding the search strategy to include multiple databases and other sources of information.
Response
We thank you for this crucial comment. Our changes are indicated in our point-by-point responses given below to the comments. All revisions in the text are indicated with red letters.

Round 2
Reviewer 3 Report
Authors have improved the manuscript sufficiently.